# Toxic Effects of Bisphenol AF Exposure on the Reproduction and Liver of Female Marine Medaka (*Oryzias melastigma*)

**DOI:** 10.3390/ani14020222

**Published:** 2024-01-10

**Authors:** Huichen Li, Jiahao Gao, Yue Liu, Yujia Ding, Yusong Guo, Zhongduo Wang, Zhongdian Dong, Ning Zhang

**Affiliations:** 1Key Laboratory of Aquaculture in South China Sea for Aquatic Economic Animal of Guangdong Higher Education Institutes, College of Fishery, Guangdong Ocean University, Zhanjiang 524088, China; lllhhhccc2000@163.com (H.L.); jhgao2001@163.com (J.G.); ly2742706669@163.com (Y.L.); 18031192772@163.com (Y.D.); ysguo@gdou.edu.cn (Y.G.); aduofa@hotmail.com (Z.W.); 2Guangdong Provincial Key Laboratory of Pathogenic Biology and Epidemiology for Aquatic Economic Animals, College of Fishery, Guangdong Ocean University, Zhanjiang 524088, China

**Keywords:** aquatic ecosystem, bisphenols, endocrine disruptor, immune response, gene expression

## Abstract

**Simple Summary:**

Bisphenol AF (BPAF) is a common environmental endocrine-disrupting chemical (EDC) that can potentially affect the growth and development of aquatic organisms. EDCs in the aqueous environment can be deposited in estuaries through runoff or diffused into the marine environment. The toxicity of the EDCs is related to exposure dose and duration. In this study, we used marine medaka as the experimental subject to dynamically observe the response of fish in terms of growth, reproduction and antioxidant immune system during BPAF exposure, providing new insights into understanding the marine ecological risk of BPAF.

**Abstract:**

In recent years, bisphenol AF (BPAF) in aquatic environments has drawn attention to its ecological risks. This study aims to investigate the toxic effects of BPAF (188.33 μg/L) exposure for 30 days on female marine medaka (*Oryzias melastigma*). On the 10th and 30th day of exposure, the toxicity was evaluated using histological analysis of the liver and ovaries and the transcription levels of genes related to the antioxidant system, immune system, and hypothalamic-pituitary-gonadal (HPG) axis. Findings revealed that (1) BPAF exposure caused vacuolation, karyopyknosis and karyolysis in the liver of marine medaka, and the toxic impact augmented with duration; (2) exposure to BPAF for 10 days facilitated the growth and maturation of primary ova, and this exposure had a comparatively inhibitory effect after 30 days; (3) exposure to BPAF resulted in a biphasic regulation of the transcriptional abundance of genes involved in antioxidant and inflammatory response (e.g., *il-8*, *cat*), with an initial up-regulation followed by down-regulation. Additionally, it disrupted the transcriptional pattern of HPG axis-related genes (e.g., *3βhsd*, *arα*). In conclusion, 188.33 μg/L BPAF can alter the expression levels of functionally related genes, impair the structural integrity of marine organisms, and pose a threat to their overall health.

## 1. Introduction

The compound bisphenol AF (BPAF) is commonly employed as a substitute for bisphenol A (BPA) and finds extensive application in the manufacturing of plastic products, including polycarbonate copolymers, food contact polymers, and electronic materials [1]. However, with the increase in the production and use of BPAF, its detection frequency in aquatic environments has been progressively rising [2,3,4,5]. The slow degradation rate and significant bioaccumulation of BPAF have garnered extensive attention due to its potential toxicity to aquatic organisms [6]. The findings of various studies have demonstrated that BPAF belongs to a category of endocrine disruptors capable of impacting the reproductive system through the estrogen receptor (ER) and aromatase (AROM) pathways, thereby stimulating the expression of *erα*, *erβ*, and *cyp19a* genes [7,8], consequently leading to an elevation in estrogen levels within the body. Exposure to BPAF can result in a diminished hatch rate of zebrafish embryos and induce physiological abnormalities in juvenile fish, such as pericardial edema, yolk sac edema, and spinal curvature, concurrently impacting the development of their nervous system [9]. Moreover, exposure to high concentrations (500 μg/L) of BPAF not only exacerbates anxiety-like behavior in adult zebrafish but may also cause neurodevelopmental defects in their offspring [10,11].

The presence of BPAF residues has been extensively detected in various aquatic environments, including inland lakes, rivers, deltaic regions characterized by the convergence of saltwater and freshwater, and oceans [12,13,14,15]. The release rate of BPA in the ocean is faster than in freshwater ecosystems [16], and the rate is positively correlated with salinity [17]. Due to the similar structure of BPAF and BPA, BPAF may pose greater threats to marine organisms than freshwater organisms. The marine medaka (*Oryzias melastigma*) presents an optimal model for researching marine environmental toxicology, owing to its small body size, clear secondary sexual characteristics, short generation cycle, high egg production, transparent embryo development, and robust salt tolerance in the range of 0–35 parts per thousand (ppt) [18,19,20,21]. Previous studies show that prolonged exposure to 73.4 and 367 μg/L BPAF decreased the transcription levels of reproductive-related genes (*arα*, *vtg* and *lhr*), impeded oogenesis, and caused liver damage in the marine medaka [22]. We also found that exposure to 200 μg/L (the actual measured concentration was 144.70 μg/L) BPAF for 70 days caused significant toxic effects on the growth, reproduction, and liver of marine medaka [18]. The toxic effects of endocrine disruptors gradually aggravate with the exposure time. To dynamically observe the toxic effects of BPAF on marine medaka, the female medaka were exposed to the same concentration (actual measured concentration was 188.33 μg/L) of BPAF for 30 days, and modifications in histological characteristics and gene transcription levels were investigated. The findings demonstrate alterations in toxicity and provide valuable insights for enhancing current theories and for assessing and regulating marine ecology by humans.

## 2. Materials and Methods

### 2.1. Ethics Statement

All experimental protocols were approved by the Animal Research and Ethics Committee of Guangdong Ocean University. The study did not involve endangered or protected species.

### 2.2. Experimental Design

BPAF (CAS, 1478-61-1, analytical grade) was purchased from Sigma–Aldrich (St. Louis, MO, USA) and dissolved in dimethyl sulfoxide (DMSO, analytical grade) to form a 20 mg/mL stock solution. Each tank in the BPAF treatment group was exposed to a concentration of 200 μg/L by adding 60 μL of BPAF stock solution. The tanks in the solvent control group were treated with 60 μL of DMSO, resulting in a volume fraction of DMSO at 0.01‰ in each tank, this concentration of DMSO is not expected to have any detrimental effects on the survival and behavior of aquatic organisms [18,22].

A total of 80 adult female marine medaka, aged 6 months, were subjected to testing for exposure to BPAF. The medaka were randomly assigned to either a solvent control group or a BPAF exposure group, with four replicates per group. Each replicate tank was housed in a glass tank containing 6 L of artificial seawater and consisted of 10 fish.

The water temperature was maintained at 26 ± 1 °C, with a salinity of 30 ± 1 ppt and a light cycle set at 14 h:10 h. The fish were fed twice a day with freshly hatched brine shrimp (*Artemia salina*). The exposure treatment spanned a duration of 30 days, during which three-quarters of the exposure solution was replaced every 48 h. The BPAF concentration was determined at both the onset and 48 h post-exposure period. On days 8, 16, and 24 of BPAF exposure, 100 mL of water samples were taken from each tank to analyze the BPAF content following the methods outlined in a previous study [18]. On days 10 and 30 of exposure, five fish from each tank were anesthetized on the ice, followed by growth parameter measurements and dissection. Terminal body length (cm) and terminal body weight (g) were measured to calculate the condition factor (K) ((body weight)/(body length)^3^ × 100). The livers and ovaries of two medaka from each tank were used for histopathological analysis. The brains, livers, ovaries, and intestines of the remaining three fish were placed in an RNA-stabilizing solution (Accurate Biotechnology (Hunan) Co., Ltd., Changsha, China) for total RNA extraction and gene transcription levels detection.

### 2.3. Histological Analysis

The liver and ovaries from eight fish were included in each treatment, and one section from the liver or ovaries was examined for each individual fish. The liver and ovaries were fixed in 4% paraformaldehyde (*w*/*v*) for 48 h and then placed in 70% alcohol for an additional 24 h. Following this, they were subjected to processing and sectioning using our previously published protocol [18]. Sections were stained with hematoxylin-eosin (HE) and photographed using a microscope (Nikon Eclipse Ni-E, Tokyo, Japan).

### 2.4. Total RNA Extraction and Gene Expression Detection

Total RNA was extracted from the brain, intestines, liver, and ovaries using *TRIzol* reagent (GDSBio, Guangzhou, China) according to the manufacturer’s protocol. RNA quality and concentration were monitored using 1% agarose gels and the NanoDrop 2000 Spectrophotometer (ThermoFisher, Waltham, MA, USA). The cDNA template was prepared following the instructions of the PrimeScript^TM^ RT reagent Kit with gDNA Eraser (Vazyme Biotech, Nanjing, China). To investigate the potential toxicity of BPAF exposure to the reproductive lipid metabolism and immune system of marine medaka, real-time fluorescence quantitative PCR (qPCR) was employed to determine the transcript levels of hypothalamic-pituitary-gonadal (HPG) axis-related genes (gonadotropin-releasing hormone receptor (*gnrhr*), follicle-stimulating hormone receptor (*fshr*), glycoprotein alpha subunit (*gthα*), steroidogenic acute regulatory protein (*star*), androgen receptor alpha (*arα*), estrogen receptor alpha (*erα*), 17-beta-hydroxysteroid dehydrogenase (*17βhsd*), 3-beta-hydroxysteroid dehydrogenase (*3βhsd*), cytochrome P450 family 11 subfamily A (*cyp11a*), cytochrome P450 family 11 subfamily B (*cyp11b*), cytochrome P450 family 19 subfamily A (*cyp19a*), estrogen receptor Beta (*erβ*), vitellogenin receptor (*vtgr*), vitellogenin 1 (*vtg1*),vitellogenin 2 (*vtg2*), luteinizing hormone receptor (*lhr*), choriogenin L (*chgl*) and choriogenin H (*chgh*)), lipid metabolism-related genes (Peroxisome proliferator-activated receptor alpha (*pparα*), peroxisome proliferator-activated receptor beta (*pparβ*), apolipoprotein C1 (*apoc1*), apolipoprotein Ba (*apoba*), cytochrome P450 family 7 subfamily A member 1 (*cyp7a1*), fatty acid synthase (*fasn*), lipoprotein lipase (*lpl*), and diacylglycerol acyltransferase 2 (*dgat2*)), and antioxidant and immune response-related genes (interleukin-8 (*il-8*), superoxide dismutase (*sod*), catalase (*cat*), cyclooxygenase 1 (*cox-1*), glutathione peroxidase (*gpx*), cyclooxygenase 2 (*cox-2*) and tumor necrosis factor (*tnf*)). Ribosomal protein S4 X-linked (*rps4x*) and actin beta-2 (*actb2*) were used as internal reference genes to correct target gene transcript levels using the qPCR method, consistent with our prior work [20]. Each sample was run in triplicate, and the transcriptional data were analyzed using the 2^−ΔΔCt^ method [23] and expressed in log_2_ form. The information and sequence of all primers in this study are given in Appendix A.

### 2.5. Statistical Analysis

Experimental data were presented as the mean ± standard error (SE). Data analysis was performed using GraphPad Prism 9 software with Student’s *t*-test. The significance of the difference was noted using asterisks (* *p* ≤ 0.05, ** *p* ≤ 0.01, *** *p* ≤ 0.001, **** *p* ≤ 0.0001).

## 3. Results

### 3.1. Measured Concentrations of BPAF in Exposure Solutions

The concentration of BPAF was measured and is documented in Appendix A. The solvent control group showed no detection of BPAF, and the treatment groups demonstrated actual BPAF concentrations (188.33 μg/L) lower than the nominal concentrations. The final exposure concentration used the average actual BPAF concentration.

### 3.2. Effect of BPAF on Marine Medaka Growth

After a 10-day exposure to BPAF, no observable changes were present in the marine medaka’s terminal body weight and length in either the BPAF-exposed group or the solvent control group (Figure 1A,B). However, following a duration of 30 days of exposure to BPAF, a marked increase in both terminal body weight and length was observed in contrast to the solvent control group (Figure 1D,E). No significant differences in the condition factor of marine medaka were observed between the exposed group and the control group after 10 and 30 days of BPAF exposure (Figure 1C,F).

### 3.3. Histopathological Analysis

As the duration of BPAF exposure increased, the degree of liver injury was aggravated in female marine medaka. After a 10-day exposure period, hepatocytes from all BPAF-exposed medaka exhibited vacuolization and karyopyknosis, accompanied by a small amount of karyolysis (Figure 2). However, following a 30-day exposure period, the severity of karyolysis increased, and intercellular spacing expanded (Figure 2). BPAF exposure also had adverse effects on medaka ovarian development and oocyte maturation. A 10-day exposure to BPAF promoted ovarian development, with the BPAF-treated group showing a higher quantity of late-stage vitellogenic/mature oocytes. At day 30, the fish in the BPAF-treated group contained a higher number of perinuclear oocytes and a decrease in late/mature oocytes compared with the control group. Additionally, the BPAF-treated group exhibited significant pathological features of follicular atresia (Figure 3 and Appendix A).

### 3.4. Effects of BPAF Exposure on the Expression of Antioxidant and Immune-Related Genes in Marine Medaka

Exposure to BPAF impacts the transcriptional levels of antioxidant and inflammatory-related genes. The transcription levels of *cox-1* and *il-8* in the liver were noticeably up-regulated after 10 days of exposure (Figure 4), as well as *cat*, *cox-1*, *il-8* and *tnf* in the intestine (Figure 5). After a period of 30 days of exposure to BPAF, the mRNA levels of *pparα*, *pparβ*, *sod*, and *cat* in the liver (Figure 4), as well as those of *pparα*, *pparβ*, *sod*, and *cox-1* in the intestine, were significantly reduced. Notably, genes (*pparβ*, *gpx*, *sod*, *cat*, *il-8*) in both the liver and intestine revealed a dynamic change of, first, up-regulation and, then, down-regulation on the 10th and 30th day of BPAF exposure. Compared with the control group, there were significant up-regulations and down-regulations of *pparβ* and *cat* (Figure 4 and Figure 5).

### 3.5. Effect of BPAF Exposure on the Expression of HPG Axis-Related Genes in Marine Medaka

Compared with the control group, the mRNA expression levels of *gnrhr*, *fshr*, *erα*, and *17βhsd* in the brain of marine medaka were significantly increased after 10 days of BPAF exposure, and the transcription levels of *lhr*, *gthα*, *cyp19b*, *arα*, and *3βhsd* showed a tendency to rise (Figure 6A). Meanwhile, in the ovaries, there was significant down-regulation in the transcript levels of *3βhsd*, *arα*, and *fshr* (Figure 6B). Significant up-regulation was observed in the *vtgr*, *vtg2*, *vtg1*, *chgl*, *chgh*, and *erβ* transcript levels in the liver (Figure 6C). After 30 days of BPAF exposure, the transcription levels of *lhr*, *fshr*, *arα*, and *3βhsd* in the brain of marine medaka were significantly up-regulated compared with those in the control group (Figure 6a). The transcript levels of *fshr* and *sta*r were significantly down-regulated in the ovaries (Figure 6b). A substantial rise in the transcript levels of *vtg2*, *vtg1*, *chgl*, and *chgh* was observed within the liver (Figure 6c). The majority of the genes measured in the brain and liver exhibited continuous up-regulation from the changes seen on exposure day 10 to day 30 (Figure 6A,a,C,c). Among the genes measured in the ovaries, the transcript level of *fshr* showed a sustained down-regulation, and the transcript levels of *3βhsd* and *arα* switched from being significantly down-regulated to showing no difference, whereas the *star* transcript levels were down-regulated in the BPAF-exposed groups (Figure 6B,b).

### 3.6. Effects of BPAF Exposure on Lipid Metabolism in Female Marine Medaka

The results showed that genes implicated in lipid metabolism undergo dynamic expression under various exposure conditions. On day 10 of exposure to BPAF, there was a significant increase in the transcript levels of *apoc1* and *dgat2*. At day 30 of the exposure, the transcription level of *dgat2* exhibited a significantly reduced trend (Figure 7). During this process, transcript levels of *cyp7a1*, *fasn*, and *lpl* showed a continuous but insignificant downward trend. The transcript level of *apoba* showed a trend of up-regulation followed by down-regulation (Figure 7).

## 4. Discussion

### 4.1. BPAF Exposure Affects the Growth of Female Marine Medaka

BPA and tetrabromobisphenol A (TBBPA) can stimulate the feeding behavior in zebrafish, accelerate their body length and weight growth, and ultimately trigger lipid accumulation, leading to obesity [24]. The hepatic fatty accumulation levels in male zebrafish were found to be significantly higher than those in females following exposure to BPS [25]. Consistent with the observed effects of BPA and its analogs on zebrafish, exposure to 188.33 μg/L BPAF resulted in a significant increase in both body length and weight of female marine medaka (Figure 1). These results may indicate that the effects of bisphenols are similar in stimulating fish growth. Future investigations should prioritize examining the impact of BPAF on zebrafish’s diet and lipid metabolism, as well as elucidating the underlying mechanisms, in order to comprehensively evaluate the potential ecological risks posed by BPAF to aquatic organisms.

### 4.2. BPAF Exposure Causes Liver Injury and Metabolic Disorders

The liver serves as the regulatory hub for fish metabolism and exhibits a high sensitivity to exogenous substances. Exposure to environmental pollutants can result in alterations to the structure and physiological properties of fish liver [26,27]. Liver toxicity resulting from exposure to bisphenols such as BPA, BPF, and BPAF has been reported in various organisms, including mice, zebrafish, and medaka [13,28,29]. In the present study, hepatocytes demonstrated vacuolation, karyopyknosis, and karyolysis (Figure 2), indicating that BPAF causes liver injury. This finding is consistent with that of previous studies that demonstrated the ability of BHPF, BPA, and BPAF to induce liver injury in mice by causing hepatocyte vacuolation and swelling [30]. The disruption of the liver’s antioxidant balance might be linked to this phenomenon. Glutathione peroxidase, mediated by SOD and GPX, functions as the liver’s primary antioxidant enzyme [31]. The exposure of marine medaka to low concentrations of BPA (1 and 10 μg/L) resulted in an up-regulation of liver antioxidant enzyme activity, whereas exposure to high levels of BPA (100 and 1000 μg/L) led to a down-regulation of the liver antioxidant enzyme activity [32]. In this study, the transcriptional levels of *sod* and *cat* showed a decreasing trend with BPAF exposure time, while *cox-1* showed an increasing trend. This pattern may indicate that the antioxidant system of medaka is still in equilibrium during the early stages of exposure to BPAF, and the balance of the antioxidant system is destroyed with prolonged exposure.

Tumor necrosis factor (TNF) is a pivotal cytokine involved in the host defense mechanism against viruses, bacteria, and parasites [33,34,35]. Cyclooxygenase (COX) is an enzyme necessary for converting arachidonic acid into prostaglandins during the inflammatory response. COX-1 regulates systemic homeostasis and maintains stable expression in the body, whereas COX-2 exhibits low basal expression but undergoes significant up-regulation upon inflammatory stimulation. TNF-α promotes the expression of COX-2 and nitric oxide synthase (iNOS) via NF-κB activation [36,37]. The activated COX-2 enhances the metabolism of arachidonic acid (AA), generating inflammatory prostaglandins (PGs), which instigate chronic inflammation [38]. Interleukin-8 (IL-8) attracts and activates neutrophils, T lymphocytes, and eosinophils and promotes the lysosomal enzyme activity and phagocytosis of neutrophils [39]. In the present study, the relative mRNA expression levels of *il-8* and *cox-1* increased in the liver after 10 days of exposure to BPAF, plausibly resulting from the body’s immune response to an external stimulus. However, at the late stage of exposure, the transcript levels of *cox-1* were consistently up-regulated, while the transcript levels of the anti-inflammatory factor *il-8* appeared to be down-regulated, which suggests a weakened immune system in the body. Combined with the histological structure of the liver, BPAF may inhibit the body’s immunity by inducing a decrease in the expression of *il-8* and *tnf*, thereby increasing the likelihood of infection. The transcription levels of the relevant genes in the intestine matched those in the liver, suggesting that exposure to BPAF has adverse effects on the viscera of marine medaka.

Fish liver is the main organ for lipid synthesis and metabolism. Diacylglycerol acyltransferase 2 (DGAT2), peroxisome proliferator-activated receptors (PPARS), and fatty acid synthase (FASN) are the key elements of triglyceride synthesis in animals [40]. Apolipoprotein Ba (APOBA), fatty acid binding protein (FABP), and apolipoprotein C-I (APOC1) are involved in lipid transport, while peroxisome proliferator-activated receptor α (PPARA) and lipoprotein lipase (LPL) are involved in lipid oxidation [41]. Exposure to BPA analogs can affect the transport and synthesis of fatty acids in zebrafish, resulting in lipid accumulation [42]. In the present study, the mRNA expression of *dgat2*, *apoc1*, and *pparβ* significantly increased after 10 days of BPAF exposure (Figure 7), indicating that BPAF enhances lipid synthesis in the medaka liver, resulting in hepatic lipid accumulation. These findings may be associated with the observed vacuolization in the liver (Figure 2). However, after a 30-day exposure to BPAF, the activity of enzymes that promote lipid synthesis (*apoc1* and *apoba*) decreased, resulting in a slowdown in lipid accumulation (Figure 7). It can be surmised that liver injury in medaka may be associated with disrupted lipid metabolism caused by the impact of BPAF on gene expression involved in fatty acid synthesis and oxidation.

### 4.3. Effect of BPAF Exposure on Transcription Levels of Genes Involved in the HPG Axis

Reproductive endocrine homeostasis in vertebrates is regulated by the HPG axis, which plays a crucial role in the normal growth and reproductive activities of fish. Therefore, the HPG axis and the liver are often easy targets for detecting the toxic effects of pollutants [43]. Gonadotropin-releasing hormones (GnRHs) play a crucial role in the reproductive neuroendocrine system being secreted from the hypothalamus, binding to the pituitary gonadotropin-releasing hormone receptor (GnRHR), activating downstream genes, and stimulating the synthesis of sex hormones [44,45]. In this study, the mRNA expression level of *gnrhr* was up-regulated in the brain after 10 and 30 days of treatment, indicating that the activity of the gonadotropin-releasing hormone receptor was stronger in the brain at this stage.

Gonadotropins FSH and LH can bind to the gonadal receptors FSHR and LHR to regulate steroidogenesis and gametogenesis in the gonads [46]. Meanwhile, steroidogenic acute regulatory protein (STAR), 17β-hydroxysteroid dehydrogenase (17βHSD), and 3β-hydroxysteroid dehydrogenase (3βHSD) control all steroidogenesis and hormone activity conversion. This study found that the mRNA expression levels of *fshr*, *lhr*, *17βhsd*, and *3βhsd* in the brain were higher after 10 and 30 days of exposure to BPAF compared with the control group. However, these genes exhibited a continuous trend of down-regulation or up-regulation in the ovaries. It is speculated that BPAF can bind to the appropriate receptors and regulate hormone-active genes, leading to the inactivation of estrogen. This results in ovarian estrogenic disorders characterized by a transient increase in vivo estrogen level, accelerated follicle maturation, and subsequent impaired development. This is consistent with the results observed in ovarian sections on days 10 and 30. These results indicate that the synthesis of sex hormones is inhibited by BPAF. An experiment in zebrafish also showed that a higher concentration (1 mg/L) of BPAF could inhibit oocyte maturation in a 28-day treatment [7,47].

Choriogenin H (ChgH) and choriogenin L (ChgL) are synthesized in the liver and incorporated into the radiation zone of mature females. The expression of Cyp19b is positively regulated by estradiol (E2) in a feedback loop [48], and the up-regulation of *cyp19b* expression indicates an increase in E2 levels, thereby promoting the synthesis of the egg precursor protein vitellogenin (VTG). In this study, the transcription levels of *chgh*, *chgl*, *vtg1*, *vtg2*, and *vtgr* in the liver were significantly up-regulated in the BPAF treatment group, indicating that BPAF can promote oocyte maturation for a short time. However, it has been shown that high concentrations of BPAF can inhibit follicle maturation in the ovaries, with a more pronounced impact as the concentration increases [22]. In the present study, ovarian histological results showed an increased incidence of follicular atresia within the ovaries after 30 days of treatment, possibly due to the effects of BPAF.

## 5. Conclusions

The present study demonstrated that exposure to 188.33 μg/L BPAF results in liver cell injuries, such as vacuolization, karyopyknosis, and karyolysis. These injuries may be caused by a disruption of the genes involved in the antioxidant system and immune system in marine medaka’s liver. Exposure to BPAF accelerated oocyte maturation at the initial stage (10 days) but inhibited oocyte maturation after 30 days of exposure. Meanwhile, BPAF also induced disruptions in the HPG axis and perturbed lipid metabolism in medaka by affecting the transcript expression levels of genes related to the HPG axis and lipid metabolism. These findings contribute to our comprehension of the aquatic toxicological effects of BPAF, suggesting its unsuitability as a viable substitute for BPA and highlighting potential threats to marine organisms.

## Figures and Tables

**Figure 1 animals-14-00222-f001:**
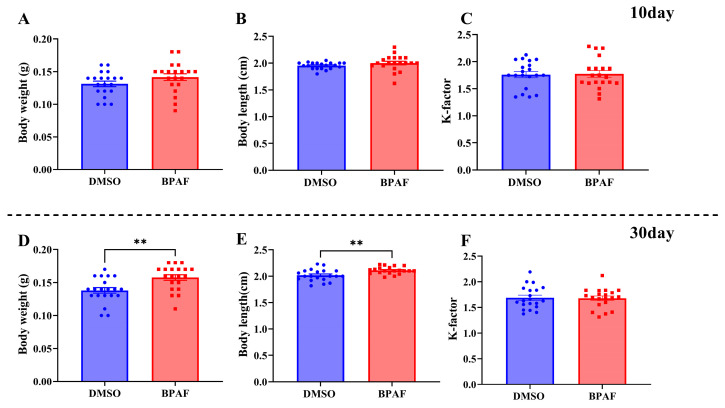
Effects of BPAF exposure for 10 and 30 days on medaka body weight (**A**,**D**), length (**B**,**E**), and condition factor (**C**,**F**). The results are represented as mean ± standard error (SE), *n* = 20. Compared with the control, statistically significant differences are shown using asterisks (** *p* ≤ 0.01).

**Figure 2 animals-14-00222-f002:**
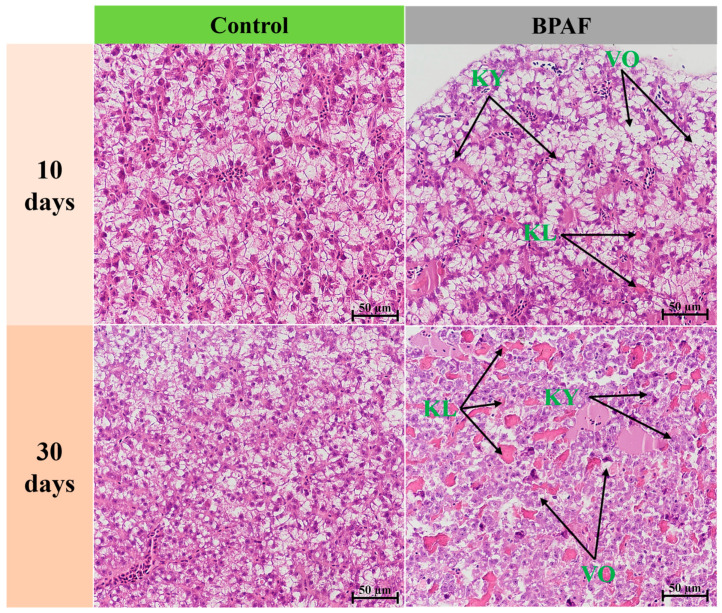
Effects of BPAF treatment on the histological structure of the liver. VO: vacuolization; KY: karyopyknosis; KL: karyolysis. Scale bar: 50 μm.

**Figure 3 animals-14-00222-f003:**
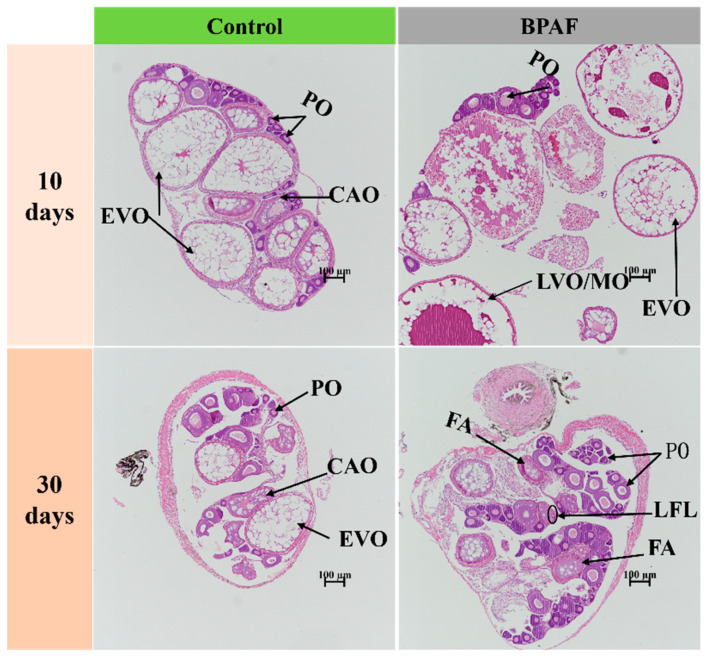
Effects of BPAF treatment on the histological structure of the ovaries. PO: perinucleolar oocytes; CAO: cortical alveolar oocytes; EVO: early vitellogenic oocytes; LVO/MO: late vitellogenic oocytes/mature oocytes; LFL: follicular lining loose (ellipse); FA: follicular atresia. Scale bar: 100 μm.

**Figure 4 animals-14-00222-f004:**
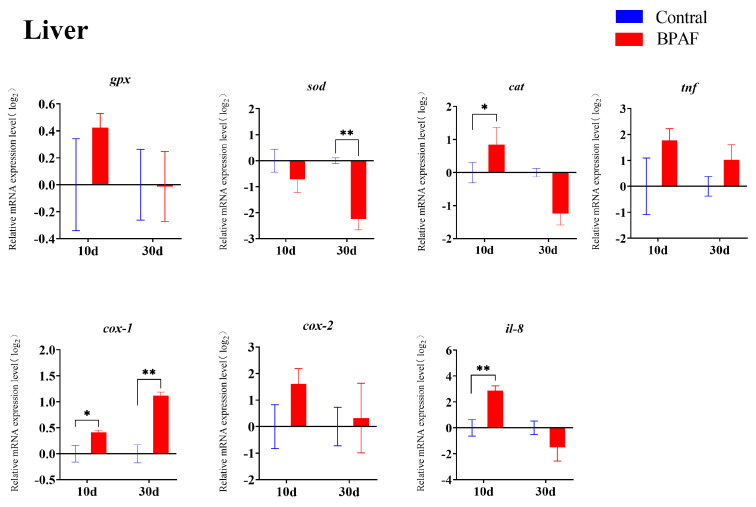
Transcript levels of genes involved in antioxidant and immune response in the liver after BPAF exposure. The results are represented as mean ± standard error (SE), *n* = 4. Compared with the solvent control group, statistically significant differences are shown using asterisks (* *p* ≤ 0.05, ** *p* ≤ 0.01).

**Figure 5 animals-14-00222-f005:**
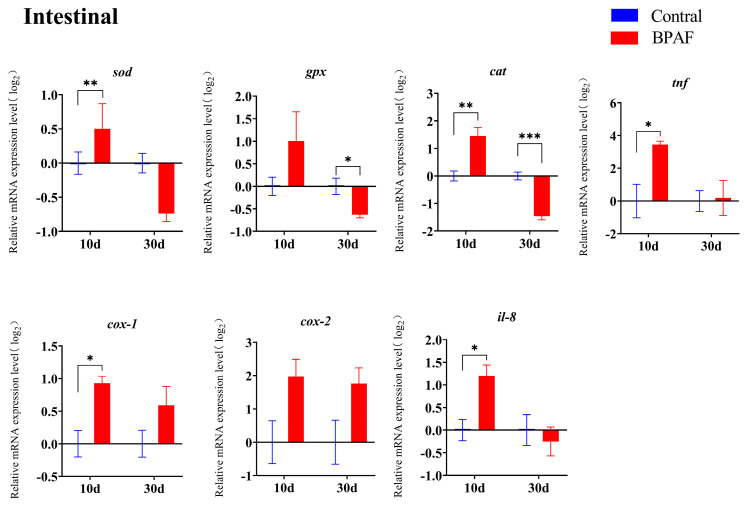
Transcription levels of genes involved in antioxidant and immune response in the intestine after BPAF exposure. The results are represented as mean ± standard error (SE), *n* = 4. Compared with the solvent control group, statistically significant differences are shown using asterisks (* *p* ≤ 0.05, ** *p* ≤ 0.01, *** *p* ≤ 0.001).

**Figure 6 animals-14-00222-f006:**
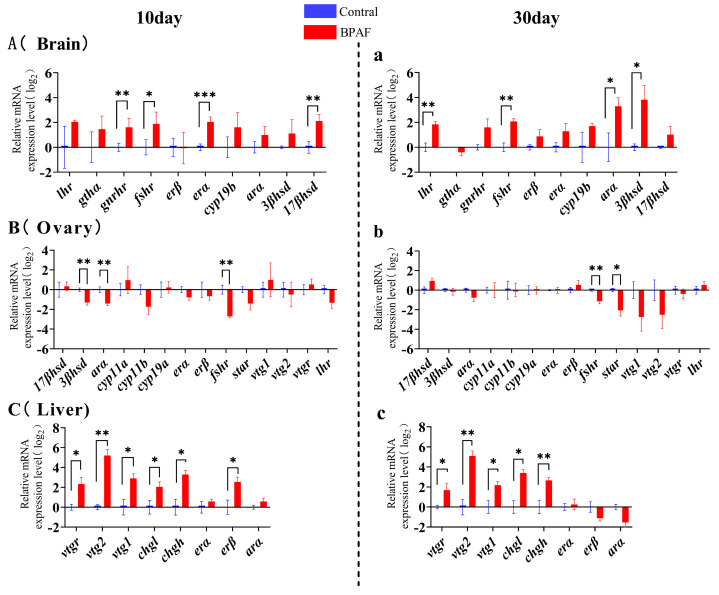
Transcriptional levels of HPG axis-related genes after BPAF exposure. Effects of BPAF exposure for 10 and 30 days on transcription levels of HPG axis-related genes in medaka’s brain (**A**,**a**), ovaries (**B**,**b**) and liver (**C**,**c**), respectively. The results are represented as mean ± standard error (SE), *n* = 4. Compared with the solvent control group, statistically significant differences are shown using asterisks (* *p* ≤ 0.05, ** *p* ≤ 0.01, *** *p* ≤ 0.001).

**Figure 7 animals-14-00222-f007:**
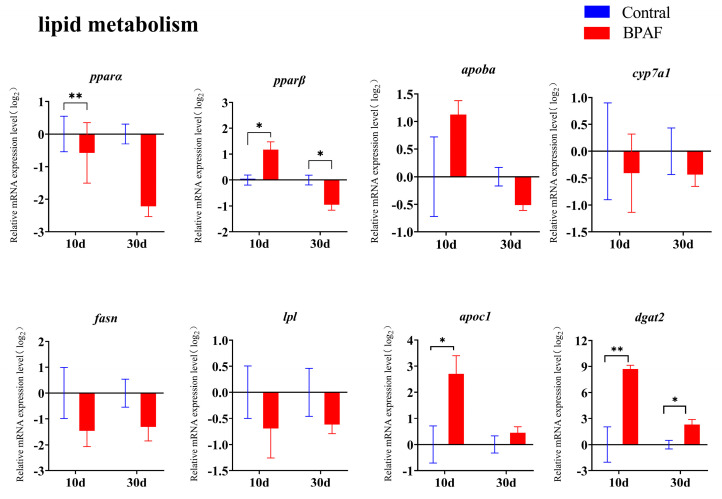
Transcriptional levels of genes related to liver lipid metabolism after BPAF exposure. The results are represented as mean ± standard error (SE), *n* = 4. Compared with the solvent control group, statistically significant differences are shown using asterisks (* *p* ≤ 0.05, ** *p* ≤ 0.01).

## Data Availability

The data presented in this study are available on request from the corresponding author.

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
