# Peer review of "Toxic Effects of Bisphenol AF Exposure on the Reproduction and Liver of Female Marine Medaka (Oryzias melastigma)"

_animals, 2024, doi:10.3390/ani14020222_

Round 1

Reviewer 1 Report (Previous Reviewer 3)

Comments and Suggestions for Authors

Dear authors, while it is true that several recommendations have been addressed, I believe that the major manuscript errors have not been tackled. Specifically, there is no mention of a true negative control, which is crucial in toxicology studies. You are exposing DMSO and BPFA, but a group not exposed to anything is missing. That would be the actual control you refer to in the figures. Furthermore, in histopathology, there is no mention of the proportion of animals presenting lesions or at least a scoring system to analyze them.

Comments on the Quality of English Language

i have no comments

Author Response

Reviewer 2 Report (Previous Reviewer 2)

Comments and Suggestions for Authors

The provided response addresses my question, yet there are still a few minor flaws in the manuscript. It is advisable to thoroughly review the manuscript and correct these minor flaws. For in-depth feedback, please consult the manuscript.

Author Response

We sincerely thank you for careful reading and valuable advice. We reviewed your comments on revisions to the manuscript and carefully revised the manuscript based on your comments. 

Reviewer 3 Report (Previous Reviewer 1)

Comments and Suggestions for Authors

none

Author Response

Thank you for your recognition of our work. We have completed the modification according to the opinions of other reviewers.

Round 2

Reviewer 1 Report (Previous Reviewer 3)

Comments and Suggestions for Authors

Dear authors, I think that the manuscrit has been improved to being publicated. However, at least, it should be reflected with bibliography in the Materials and Methods, that DMSO does not cause any type of adverse effect.

This manuscript is a resubmission of an earlier submission. The following is a list of the peer review reports and author responses from that submission.

Round 1

Reviewer 1 Report

Comments and Suggestions for Authors

Dear authors,

The study investigated the toxic effects of BFAF on the liver and reproduction of female marine medeka. Histological and molecular methods were used in the study. The work was prepared very carelessly. The materials and methods and findings section is confusing. Additionally, I do not recommend that the manuscript be accepted into the journal "Animals" for the reasons I stated below.

1-The English of the manuscript should be checked by an expert (For example, Fresh brine shrimp were fed twice daily.).

2- In the manuscript, figures should be cited within the text.

3- How was the applied BFAF dose (188,33 µm/L) chosen in this study? Was the lethal dose of BFAF for marine medaka determined in this study?

4-There is a graph of growth factors in Figure 1. There is no information about growth factors in the material method. Could this be the condition factor?

5- Standard error was used in the "statistical analysis" section, but it is stated as a standard deviation in the results. Please check it.

6-In the results, the materials and methods of the intestinal analysis were not mentioned.

7- The symptoms indicated in Figure 2 are not clear. Although the authors refer to it as arrows, the symbols are triangles. and it is not clear which parts it shows. Use the arrow just like in Figure 3.

8- The localities of the bars used in the histological figures are inappropriate and disproportionate.

9- Could it be the melanomacrophage center shown with the red arrow in Figure 2? Please examine carefully.

Comments on the Quality of English Language

none

Reviewer 2 Report

Comments and Suggestions for Authors

This study investigated the toxic effects of bisphenol AF (BPAF) exposure on female marine medaka fish. Fish were exposed to 188.33 μg/L BPAF for 10 or 30 days. Toxicity was evaluated by analyzing liver and ovary histology, measuring growth parameters, and quantifying expression of genes related to the hypothalamic-pituitary-gonadal (HPG) axis, immune function, and lipid metabolism. These results provide new insights for understanding the Marine ecological risk of BPAF. The manuscript has been well written. However, there are a few revisions that are necessary before the manuscript is considered for publication.

abstract

1. Condense the methods into 1-2 sentences summarizing sample size, exposure conditions, endpoints.

2. Specify the BPAF exposure concentration and duration in the results.

3. Add a conclusion sentence emphasizing BPAF’s apparent toxicity to marine fish.

4. Remove redundant phrases and details that are not needed in the abstract.

introduction

line 55-56: Rephrasing to clarify that BPAF residues have been detected globally in various water bodies.

Line 101: More specifics on the sample sizes for each analysis type.

Line 123 the − ΔΔCt method? Corrected.

discussion

line 293-295: Explaining the wider significance of impacts on the HPG axis.

Line 337: Clarifying the hypothesized dynamics of estrogen levels.

Reviewer 3 Report

Comments and Suggestions for Authors

Dear authors, your manuscript discusses BPAF, an analogue of BPA, which has also been observed to have adverse effects. Endocrine disruptors are substances of great concern today. Currently, several fish species, such as medaka and zebrafish, are used to assess the toxicity of these compounds. Regarding your work, I must say that I have found several serious methodological errors that need to be reviewed and clarified. In general, the materials and methods section is quite confusing. I am also concerned about the lack of a negative control, as it is necessary in this type of study. Groups, sample collection, and other details should be clarified. It's advisable to add the sample size (n) to each graph. Below, I provide my review.

The simple summary and the summary seem well done to me.

In the keywords, replace "HPG" with "endocrine disruptor." It's advisable not to repeat words used in the title such as "marine medaka" or "BPAF." Use terms like "aquatic ecosystem" or "bisphenols," for example.

Line 63. Use units to indicate the salt tolerance, for example, µS/cm.

Lines 46 and 67. Don't be too categorical about the toxicity of BPAF being greater than that of BPA, as this could depend on certain factors such as dosage or exposure time.

In general terms, the introduction is correct.

Regarding the beginning of the materials and methods section, I would clearly define a section for the animals, another for the chemicals used, and another for the experimental design. In each of these sections, I would provide detailed information on relevant aspects. For instance, in the animals section, I would specify their source, strain (if applicable), and the monitoring of their environmental conditions in the facilities. The way it is expressed in the text is confusing. Additionally, some relevant information is missing. Why were those doses of BPAF used? Why was DMSO used as the solvent and not another substance? This should be well justified in any toxicology study. Furthermore, in both cases, you are exposing subjects to two chemicals, BPAF and DMSO. Why is there no group that is not exposed to anything?

The sentence from line 98 is repeated in line 103.

Line 96- How were the fish sacrificed?

What I understand in the materials and methods is that you use only one dose of BPA with multiple replicates and compare it to the solvent control (I would like to emphasize again that there is no negative control). My question is, why do you use an ANOVA and a post hoc test? Wouldn't it be simpler to just use the Student's t-test? Furthermore, it does not indicate that normality and equality of variances have been assessed. Please review the statistical section. I would like to reiterate that it is necessary to clarify the experimental procedure itself.

140. “Error! Reference source not found.Error! Reference source not found”.  I assume this is a minor manuscript error. Please review it.

Figure 1. The letters and numbers in Figure 1 appear very small. I am wondering the following: if you have taken 5 fish from each tank, you should see 20 data points in each column (4 replicates * 5 animals) on the graph as individual values for each fish. For example, in graph A, there are only 11 data points in DMSO column; please clarify this.

Line 151. Histophatological analysis. The photos and descriptions seem accurate to me. My question is, you have used 2 animals out of every 10, so you would have 8 animals to analyze. In what proportion did these alterations occur? I understand that conducting morphometry is complex, but have you used a score at least?

Line 164. The scales are not visible in Figure 2.

The figures are not referenced in the text due to the error mentioned earlier, correct?

Line 187. It's impossible to know which gene is being compared in Figure 4. Split the figure into liver and intestine. Revise statistical analysis.

Except for the fact that it's impossible to read some of the figures, the text seems correct in sections 3.4, 3.5, and 3.6.

Line 237. You can relate this to the fact that this effect has been observed with BPA.

Line 247. Why do you use Oryzias latipes instead of common medaka? Be consistent.

Correct some grammatical errors in the discussion. I believe this part of the work is well executed. However, it is based on results and a methodology that should be reviewed and clarified.

Comments on the Quality of English Language

I have no comments about this.